# Tonic inhibition of the chloride/proton antiporter ClC-7 by PI(3,5)P2 is crucial for lysosomal pH maintenance

Xavier Leray[†], Jacob K Hilton[†], Kamsi Nwangwu, Alissa Becerril, Vedrana Mikusevic, Gabriel Fitzgerald, Anowarul Amin, Mary R Weston, Joseph A Mindell*

Membrane Transport Biophysics Section, National Institute of Neurological Disorders and Stroke, Bethesda, United States

**Abstract** The acidic luminal pH of lysosomes, maintained within a narrow range, is essential for proper degrative function of the organelle and is generated by the action of a V-type H+ ATPase, but other pathways for ion movement are required to dissipate the voltage generated by this process. ClC-7, a Cl-/H+ antiporter responsible for lysosomal Cl- permeability, is a candidate to contribute to the acidification process as part of this 'counterion pathway' The signaling lipid PI(3,5)P2 modulates lysosomal dynamics, including by regulating lysosomal ion channels, raising the possibility that it could contribute to lysosomal pH regulation. Here, we demonstrate that depleting PI(3,5)P2 by inhibiting the kinase PIKfyve causes lysosomal hyperacidification, primarily via an effect on ClC-7. We further show that PI(3,5)P2 directly inhibits ClC-7 transport and that this inhibition is eliminated in a disease-causing gain-of-function ClC-7 mutation. Together, these observations suggest an intimate role for ClC-7 in lysosomal pH regulation.

*For correspondence:
mindellj@ninds.nih.gov

[†]These authors contributed equally to this work

**Competing interest:** The authors declare that no competing interests exist.

## Editor's evaluation

This article will be of broad interest to readers in the field of lysosomal biology. It demonstrates that the lysosomal phosphoinositide PI(3,5)P2 tonically inhibits the CLC-7/Ostm1 chloride/proton antiporter. Relief of this inhibition leads to lysosome overacidification and enlargement and can explain the phenotype of a disease-causing 'gain-of-function' mutation in CLC-7. Together, these results suggest that the PI(3,5)P2/CLC-7 interaction may play a regulatory role in lysosome homeostasis.

## Introduction

Lysosomes are acidic organelles that participate in cellular defense and homeostasis by clearing and recycling intra- and extracellular components and informing the cell of its stress and metabolic status (*Levine et al., 2011*; *Saftig and Klumperman, 2009*; *Settembre et al., 2013*). Their acidity provides an optimal environment for luminal enzyme activity and is a source of electrochemical energy for lysosomal transporters (*Müller et al., 2012*; *Sagné and Gasnier, 2008*). Two main systems act in tandem to maintain a set point value ranging between 4.5 and 5.0 pH units: a V-ATPase that pumps and concentrates protons inside the lysosomal lumen using ATP hydrolysis, and a counterion pathway that dissipates the membrane electrical potential generated by V-ATPase activity (*Ishida et al., 2013*; *Mindell, 2012*). We currently do not know if the counterion pathway is a single system or a set of channels and transporters with different ion selectivities. One candidate is the lysosomal transporter ClC-7 of the CLC family (*Kornak et al., 2001*): it is expressed widely, located predominantly on late-endosome/lysosomal membranes, exports 1H+ to import 2Cl- per transport cycle, and is the

major lysosomal chloride conductance (*Graves et al., 2008*; *Lange et al., 2006*; *Leisle et al., 2011*). Although ClC-7's role as a counterion pathway is still debated (*Graves et al., 2008*; *Ishida et al., 2013*; *Jentsch and Pusch, 2018*; *Kasper et al., 2005*; *Kornak et al., 2001*; *Mindell, 2012*; *Steinberg et al., 2010*; *Weinert et al., 2014*), new work on patients with a novel ClC-7 disease-causing mutation strongly supports this hypothesis, with a gain of function of the transporter resulting in lysosomal hyperacidification (*Nicoli et al., 2019*).

Recent work established a connection between ClC-7 and the phospholipid kinase PIKfyve (*Gayle et al., 2017*). PIKfyve synthesizes phosphatidylinositol-3,5-biphosphate (PI(3,5)P2) from PI3P at the lysosomal membrane and participates in the maturation and trafficking of endosomes and lysosomes (*Hasegawa et al., 2017*; *Jin et al., 2016*). While it is well established that pharmacological inhibition or gene disruption of PIKfyve depletes PI(3,5)P2, alters endolysosomal trafficking, and leads to generation of large cytoplasmic vacuoles (*Bissig et al., 2017*; *Chow et al., 2007*; *Choy et al., 2018*; *Compton et al., 2016*; *Ikonomov et al., 2001*; *Sharma et al., 2019*; *Yamamoto et al., 1995*; *Zolov et al., 2012*), relatively few studies have examined the effects of PIKfyve inhibition on lysosomal pH. In these studies, PIKfyve inhibition had no effect in some cell types but did affect the pH in others, depending on the inhibitor used and the measurement method (*Ho et al., 2015*; *Sharma et al., 2019*). In this context, it was surprising that *ClCN7* and *OSTM1* (ClC-7's β-subunit) were among only four hits in a global CRISPR screen targeting 19,050 protein-coding genes to confer apilimod resistance, and the only two genes that, upon CRISPR knockdown, reduced PIKfyve-inhibitor-induced vacuole formation (*Gayle et al., 2017*). Given the potential role of ClC-7 in lysosomal acidification, these results suggest that PIKfyve might regulate lysosomal pH by modulating ClC-7 activity, and might thereby tune lysosomal traffic. Consistent with this hypothesis, PI(3,5)P2 inhibits ClC-a, a CLC anion/H+ exchanger that contributes to the vacuolar acidification of plant cells, equivalent to lysosomes in animal cells (*Carpaneto et al., 2017*).

Here, using ratiometric measurements of lysosomal pH in cells treated with the PIKfyve inhibitor apilimod, we first explored the effects of PIKfyve inhibition on lysosomal pH and size. We then assessed the relationship between lysosomal pH and vacuole formation. Finally, we evaluated the role of ClC-7 in this process using both pH measurements of lysosomes from ClC-7 knockout cells and patch-clamp electrophysiology of ClC-7 trafficked to the plasma membrane. Moreover, we link these effects to a recently identified rare disease mutation, Y715C, that causes enlarged vacuoles and lysosomal hyperacidification, similar to the effects of apilimod, providing insight into the mechanism of PI(3,5)P2 action on ClC-7. Altogether, our data reveal that PI(3,5)P2 inhibition of ClC-7 provides a brake for the lysosomal acidification mechanism, preventing overacidification by the V-type ATPase. Our results further indicate that vacuole formation is not tightly coupled to lysosomal pH and that vacuoles can form even when normal lysosomal pH is perturbed. Finally, we demonstrate that the mechanisms of PI(3,5)P2 inhibition of ClC-7 and of the recently described disease-causing mutation Y715C are tightly intertwined.

## Results

### Lysosomes swollen by PIKfyve inhibition are hyperacidic

We initially sought to determine the effect of PIKfyve inhibition on lysosomal pH using the osteosarcoma cell line U2OS; U2OS lysosomes are easily loaded with pH-sensing dyes and are distributed in the cells sparsely enough to image quantitatively and robustly. As our pH probe, we used Oregon Green 488-dextran (OG), a ratiometric pH-sensitive dye with a pKa of 4.8, well suited for the lysosomal pH range and which is delivered effectively to lysosomes via fluid-phase endocytosis (*DiCiccio and Steinberg, 2011*).

We loaded endolysosomes and lysosomes, hereafter referred to as lysosomes, with OG using standard methods with an overnight pulse of OG488-dextran followed by a 4 hr chase in dye-free media to allow the dye to accumulate in lysosomes. Most OG+positive vesicles are lysosome-related compartments, as shown by staining with both cathepsin B and LysoTracker (*Figure 1—figure supplement 1*). After delivering OG fully to lysosomes, we treated cells for 3 hr with 100 nM of the PIKfyve inhibitor apilimod or its vehicle (0.25% DMSO), enough to induce visible cytoplasmic vacuoles (*Figure 1A and B*). The OG-containing vacuoles seen in these cells are enlarged lysosomes (*Bissig et al., 2017*; *Choy et al., 2018*; *Sharma et al., 2019*). We measured lysosomal pH using excitation-ratio imaging,

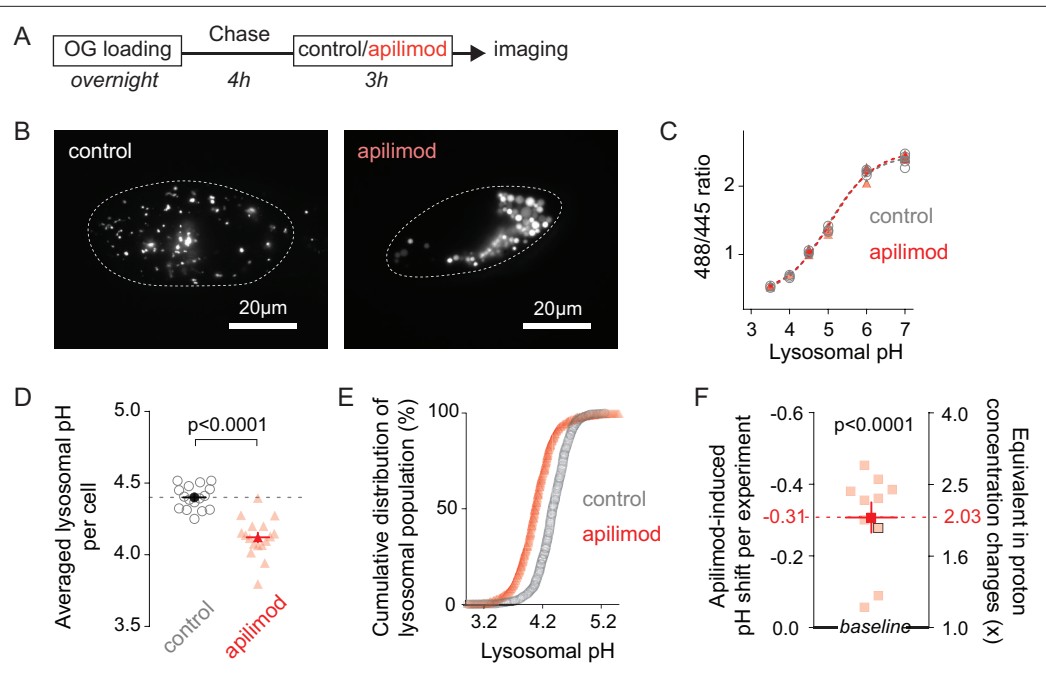

**Figure 1.** Lysosomes swollen by PIKfyve inhibition are hyperacidic. (**A**) Protocol timeline. U2OS cells were 'lysosome-loaded' with Oregon Green 488 dextran (OG loading) and treated for 3 hr with PIKfyve inhibitor apilimod (100 nM, red) or its vehicle (0.25% DMSO, control) before imaging. (**B–E**) A representative experiment. (**B**) Images of cells acquired by 445 nm laser excitation. Bright objects represent OG-positive lysosomes in control (left) versus apilimod (right) conditions. Dotted lines delineate cell outlines. (**C**) pH calibration curves obtained in control (gray empty circle) or apilimod (red triangle) conditions. Each symbol represents the averaged lysosomal 488/445 ratio of one cell (four cells per condition). (**D**) Lysosomal pH measured in control (left, gray empty circles, 16 cells, 4.40 ± 0.02) versus 100 μM apilimod (red triangles, 17 cells, 4.12 ± 0.03) for an individual experiment. Dark symbols represent averages over all cells in the experiment. Each pale symbol represents the averaged lysosomal pH of one cell. Unpaired $t$-test: p<0.0001. (**E**) Cumulative distribution of individual lysosomal pH for control (gray empty circle, 16 cells, 732 lysosomes) versus apilimod (red triangle, 17 cells, 628 lysosomes) conditions from the same experiment as in *Figure 1D*. Each symbol represents the fraction of lysosomes having a pH value below that represented in the abscissa. (**F**) Apilimod-induced pH shifts (–0.31 ± 0.04) from multiple independent experiments and corresponding fold changes in proton concentration (2.03 ± 0.17); 10 experiments (8–17 cells per condition in each experiment). Proton concentration change ($[H^+]_{change}$) was calculated from apilimod-induced pH shift ($\Delta pH$) using the following relation: $[H^+]_{change} = 10^{-\Delta pH}$. Paired $t$-test: p<0,0001. Dark symbols represent averages over all independent experiments. Each pale symbol represents the averaged lysosomal pH from multiple cells in one experiment. The dot delineated by a black box corresponds to the experiment presented in panels (**B–E**). Data are displayed as mean ± SEM.

The online version of this article includes the following figure supplement(s) for figure 1:

**Figure supplement 1.** Oregon Green 488-dextran loaded vesicles represent most of the late endocytic and lysosomal compartments.

**Figure supplement 2.** PIKfyve inhibition causes hyperacidification in neonatal fibroblasts and SUDHL cells.

including full pH calibration curves in every experiment. Though apilimod causes dramatic changes in lysosome size, overlap between pH calibration curves in control and apilimod-treated conditions shows that vacuole properties do not interfere with pH measurement (*Figure 1C*). A representative experiment reveals a substantial shift in lysosomal pH in cells treated with apilimod compared with control (*Figure 1D*), further supported by averaged results from multiple independent experiments (*Figure 1F*). In control conditions, the lysosomal pH is 4.32 ± 0.04 (mean ± SEM). We find that, in addition to its well-known effects on size, apilimod treatment induces robust lysosomal hyperacidification, reaching an average pH of 4.02 ± 0.05. This decrease of 0.31 pH unit corresponds to a doubling of the free proton concentration of the lysosomal lumen compared to control (*Figure 1F*). The global shift of the cumulative distribution of individual lysosomal pHs reveals that apilimod-induced lysosomal

hyperacidification is homogeneous and not restricted to a subset of lysosomes (*Figure 1E*). Similar measurements in neonatal fibroblasts and SUDHL B-cell lymphoma cells confirmed that their lysosomes also hyperacidify when treated with apilimod (*Figure 1—figure supplement 2*). Taken together, these results establish that acute pharmacological inhibition of PIKfyve indeed affects lysosomal pH, surprisingly causing hyperacidification. Well-documented hyperacidification is rare in lysosomal manipulation or disease, though we recently reported a disease-causing ClC-7 gain-of-function mutation with this effect (*Nicoli et al., 2019*).

## Lysosomal swelling and hyperacidification are independent events

In cultured cells, pharmaceutical inhibition of PIKfyve depletes PI(3,5)P2 on the time scale of minutes (*Zolov et al., 2012*). Subsequently, lysosomes increase in size steadily over time (*Choy et al., 2018*), an effect that reverses over several hours after drug removal (*Choy et al., 2018*; *Bissig et al., 2017*). We analyzed the time course of PIKfyve-induced lysosomal swelling and hyperacidification to investigate the relationship between the two effects.

Measuring lysosomal pH and size in OG-loaded U2OS cells treated with 100 nM apilimod for 30 min, 1 hr, 3 hr, and 24 hr revealed that, as previously reported, the initial appearance of vacuoles is rapid (*Figure 2A, B, and D*; *Choy et al., 2018*). However, whereas lysosome hyperacidification is complete (stabilizing at 0.27 ± 0.05 pH below control) during the first hour of treatment, the organelles continue enlarging over the next 24 hr (*Figure 2C*). Treatment with another recently identified PIKfyve inhibitor, WX8 (1 µM) (*Sharma et al., 2019*), recapitulates the effects of apilimod, ruling out an off-target effect (*Figure 2E*). Plotting pH versus size of individual lysosomes during hyperacidification (30 min and 1 hr) revealed no detectable correlation (*Figure 2—figure supplement 1*). After apilimod removal, lysosomes quickly begin to shrink via tubulation events (*Figure 2F and G*, orange box, and *Figure 2I*) but lysosomal pH starts to recover only after 2 hr (when lysosomal size is almost back to normal *Figure 2H and I*), fully recovering more than 6 hr after apilimod washout (*Figure 2H*).

Thus, both the kinetics and extent of the changes in lysosomal size and pH during and after apilimod treatment are quite distinct, suggesting that the processes underlying these changes are at least partially independent.

## Lysosomal hyperacidification is not required for vacuole formation

Our results regarding the kinetics of apilimod-induced lysosomal pH and size changes suggest that these two effects may occur through different mechanisms (*Figure 2H and I*). To test this possibility, we evaluated whether alkalinizing lysosomal compartments could prevent vacuole formation. This question was previously addressed in part using the V-ATPase inhibitor BafA1 and the lysosomotrophic agents chloroquine and NH₄Cl in the monkey fibroblast COS7 cell line (*Compton et al., 2016*). Although V-ATPase inhibition in that study clearly prevents vacuole formation, the pH-modifying reagents NH₄Cl and chloroquine used in that paper were added 40 min after pharmacological inhibition of PIKfyve; given our results, this changes the pH well after the development of lysosomal hyperacidification and probably too late to affect early pH-dependent processes (*Figure 2C*).

To determine the influence of lysosomal pH *early* in the process of apilimod-induced enlargement, we manipulated the pH by adding either 12 µM of chloroquine, 100 nM of BafA1, or a vehicle (water/DMSO) to OG-loaded U2OS cells 30 min *prior to* and *throughout* 3 hr 100 nM apilimod treatment (*Figure 3A*), then imaged cells to quantify lysosomal pH and size (*Figure 3B*). 30-minute treatment with 12 µM chloroquine is long enough to alkalinize lysosomes to ~pH 5.5–6.0 (*Figure 3*, *Figure 3—figure supplement 1*), a range where OG is still sensitive to pH variations (*Figure 1C*). As previously reported, inhibition of V-ATPase with BafA1 prevents vacuole formation (*Compton et al., 2016*; *Sharma et al., 2019*; *Figure 3D*). Conversely, vacuoles still form with chloroquine pretreatment (*Figure 3D*). To distinguish the effect of pH from that of direct V-ATPase inhibition, we added 25 µM chloroquine to raise the lysosomal pH to above the value resulting from bafilomycin treatment. We still observed the same level of vacuolization upon apilimod treatment, ruling out the pH change due to V-ATPase inhibition as the cause (data not shown). Interestingly, even after alkalinization with chloroquine, apilimod exposure significantly lowered lysosomal pH (*Figure 3C*; 5.63 ± 0.13 SEM [untreated] versus 5.22 ± 0.10 SEM [apilimod]; paired *t*-test, p=0.02). Both hyperacidification and vacuole formation are entirely lost when the V-ATPase is inhibited (*Figure 3C*; 6.26 ± 0.20 [untreated] versus 6.25 ± 0.23 [apilimod]), although OG is poorly sensitive to pH changes at this almost neutral pH (*Figure 1C*).

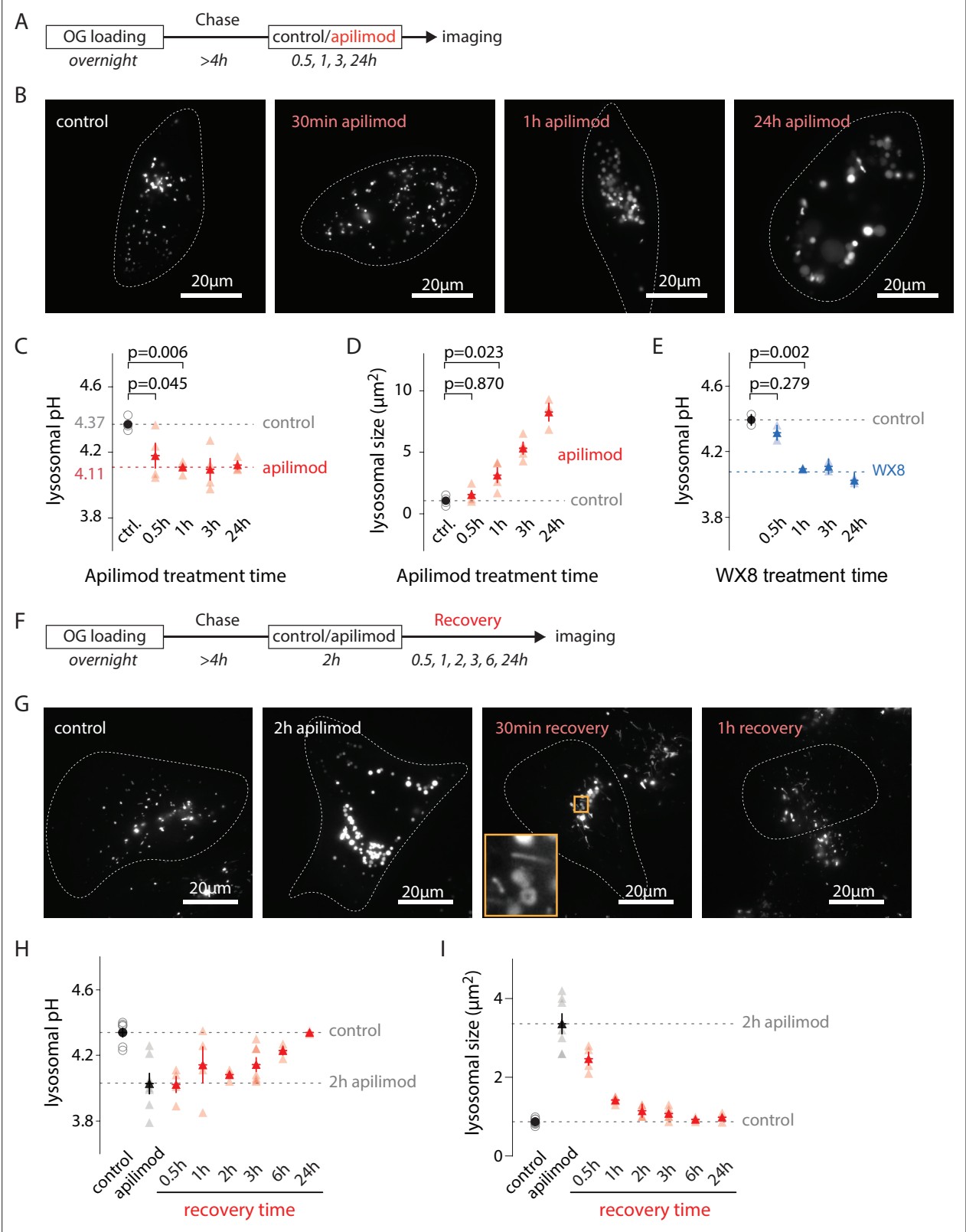

**Figure 2.** Evolution of lysosomal pH and size during PIKfyve inhibition and recovery. (**A**) Protocol timeline to analyze the evolution of lysosomal pH and size during PIKfyve inhibition. U2OS cells were 'lysosome-loaded' with Oregon Green 488-dextran (OG) then treated for 30 min, 1 hr, 3 hr, or 24 hr with apilimod (100 nM, red) or its vehicle (0.25% DMSO, control) before imaging. (**B**) Representative images of cells acquired by 445 nm laser excitation. Bright objects represent OG-positive lysosomes. Dotted lines delineate cell outlines. (**C, D**) Comparison of lysosomal pH (**C**) and size (**D**) in control (gray

*Figure 2 continued on next page*

*Figure 2 continued*

empty circle) and after different apilimod treatment times (red triangle). Each symbol represents the averaged lysosomal pH of one experiment (four independent experiments, 10–15 cells per condition per experiment). Red dashed line in (**C**) represents the average pH value of 1–24 hr apilimod time points. p-Values were obtained from one-way ANOVA Dunnett's multiple-comparisons test. (**E**) Comparison of lysosomal pH in control condition (gray empty circle) and after different treatment times with the PIKfyve inhibitor WX8 (1 μM, blue triangle). Each symbol represents the averaged lysosomal pH of one experiment (two experiments, 10–13 cells per condition per experiment). Blue dashed line represents the average value of 1–24 hr WX8 time points. (**F**) Protocol timeline to analyze the recovery of lysosomal pH and size after washout of apilimod. U2OS cells were 'lysosome-loaded' with OG, treated 2 hr with 100 nM apilimod or its vehicle (0.25% DMSO, control) to induce lysosomal hyperacidification and swelling. Apilimod was subsequently washed out with fresh media and imaged 30 min, 1 hr, 3 hr, 6 hr, and 24 hr after the washout. (**G**) Representative images of cells acquired using 445 nm laser excitation. Bright objects represent OG-positive lysosomes. Dotted lines delineate cell outlines. The orange box at 30 min time point highlights lysosomal tubulation events. (**H, I**) Comparison of lysosomal pH (**H**) and size (**I**) in control condition (gray empty circle), after 2 hr apilimod treatment (black triangle) and at different times after apilimod washout (red triangle). Each symbol represents the average lysosomal pH of one experiment (3–7 experiments, 8–15 cells per condition per experiment).

The online version of this article includes the following figure supplement(s) for figure 2:

**Figure supplement 1.** Under PIKfyve inhibition, lysosomal pH does not correlate with lysosomal size.

Taken together, these results prove that the early hyperacidification of lysosomes induced by PIKfyve inhibition is not required for vacuole formation but confirm that V-ATPase activity is necessary for the process.

## Increasing PI(3,5)P2 levels do not affect lysosomal pH

PIKfyve inhibition, and consequent PI(3,5)P2 *depletion*, hyperacidifies lysosomes to a new, lower, pH setpoint (***Figure 2C***). We also tested whether *increasing* PI(3,5)P2 levels influenced lysosomal pH using a hyperactive mutant form of PIKfyve (PIKfyve[KYA]), which raises PI(3,5)P2 levels by several fold in mouse hippocampal cultured neurons when overexpressed (***McCartney et al., 2014b***). As a negative control, we overexpressed WT PIKfyve (which does not change PI(3,5)P2 levels) (***McCartney et al., 2014b***).

We measured lysosomal pH of OG-loaded U2OS cells 24 hr after transfection with either PIKfyve[KYA] or PIKfyve (***Figure 4A and B***). Because both PIKfyve versions were untagged, we used cotransfected mCherry as a marker for transfection (***Figure 4C***); since co-transfected plasmids usually are delivered to the same cells, the mCherry-positive population must also be enriched in cells expressing the PIKfyve constructs. Note that mCherry expression does not impact lysosomal pH (***Figure 4D***). Cells coexpressing mCherry and PIKfyve[KYA] show similar lysosomal pH and size compared to controls (***Figure 4D and E***). Thus, while lowering PI(3,5)P2 results in a pH change, raising it does not. Perhaps the effector(s) downstream of the lipid in the pathway that affects pH is already saturated with the lipid under normal cellular conditions in U2OS cells.

## ClC-7 is involved in both lysosomal hyperacidification and size regulation

Recently, Gayle et al. observed that the loss of either ClC-7 or its obligate β-subunit, Ostm1, provides resistance against vacuole formation induced by apilimod treatment in B-cell non-Hodgkin's lymphoma (***Gayle et al., 2017***). Since ClC-7 has been hypothesized to facilitate lysosomal acidification (***Graves et al., 2008***; ***Ishida et al., 2013***; ***Mindell, 2012***), and as apilimod shifts lysosomal pH to a new setpoint (***Figure 2C***), we analyzed the impact of *CLCN7* knockout on PIKfyve inhibition, generating a U2OS *CLCN7* knockout cell line (ClC-7 KO) using CRISPR/Cas9 (***Shalem et al., 2014***). Without good available antibodies for ClC-7, we validated *CLCN7* allele deletion both at genomic and transcript levels (***Figure 5A***, ***Figure 5—figure supplement 1***). This ClC-7 KO cell line has two different *CLCN7* alleles with 64 or 118 bp frameshift deletions in EXON1 (***Figure 5B***). No wildtype allele remains.

To probe for effects of knocking out ClC-7 on apilimod sensitivity, we measured lysosomal pH in WT and ClC-7 KO U2OS cells (100 nM apilimod, 3 hr; ***Figure 5C and D***). Overlap of both pH calibration curves and cumulative distribution plots of individual lysosomal pH for WT and KO lysosomes demonstrates that knocking out ClC-7 affects neither lysosomal OG loading nor trans-membrane pH equilibration (***Figure 5—figure supplement 2***). Consistent with previous reports for several cell types in primary culture (***Kasper et al., 2005***; ***Lange et al., 2006***; ***Weinert et al., 2010***), knocking out ClC-7 in untreated U2OS cells does not alter lysosomal pH (***Figure 5G***). The size of untreated

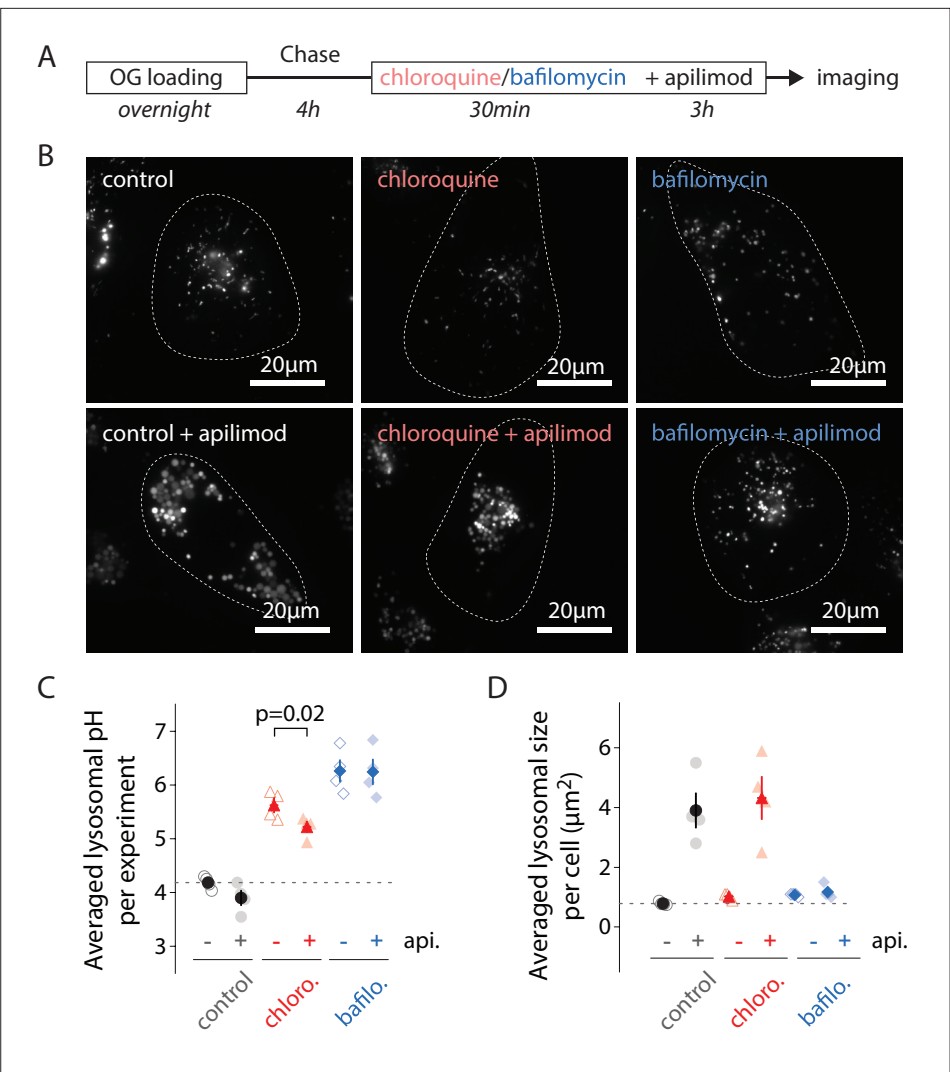

**Figure 3.** Alkilinazing lysosomal pH does not prevent lysosomal swelling. (**A**) Protocol timeline. U2OS cells were 'lysosome-loaded' with Oregon Green 488-dextran (OG) and pretreated 30 min with chloroquine (12 μM, red), BafA1 (100 nM, blue), or a vehicle (0.31% DMSO, gray), before 3 hr treatment with apilimod (100 nM, bottom images) or a vehicle (control, DMSO 0.31%, upper images). (**B**) Representative images of cells acquired by 445 nm laser excitation. Bright objects represent OG-positive lysosomes. Dotted lines delineate cell outlines. (**C, D**) Comparison of lysosomal pH (**C**) or size (**D**) in control (gray), chloroquine (chloro., red), and BafA1 (bafilo., blue) conditions during untreated (api. -, empty symbols) versus apilimod treated (api. +, filled symbols) conditions. Each symbol represents the averaged lysosomal pH or size of one experiment (four experiments; 8–11 cells per condition). In (**C**), apilimod induces a lysosomal pH shift to more acidic value in chloroquine condition (paired *t*-test, p=0.02).

The online version of this article includes the following figure supplement(s) for figure 3:

**Figure supplement 1.** 30-minute treatment with chloroquine is sufficient to induce alkalization of endo/lysosomal compartments.

CLC-7 KO lysosomes is also unchanged from WT (*Figure 5F*). Strikingly, however, knocking out ClC-7 strongly diminishes apilimod-induced lysosomal hyperacidification: the ΔpH is only about 0.13 units in ClC-7 KO cells compared to 0.31 units in WT cells (*Figure 5*, representative experiment in *Figure 5E*; average of multiple independent experiments in *Figure 5H*; 0.31 pH units ± 0.04 SEM, WT; versus 0.13 pH units ± 0.04 SEM, ClC-7; p=0.0119). Similarly, lysosome swelling still occurs upon apilimod treatment in ClC-7 KO cells, but to a lesser extent (*Figure 5*, representative experiment in *Figure 5F*;

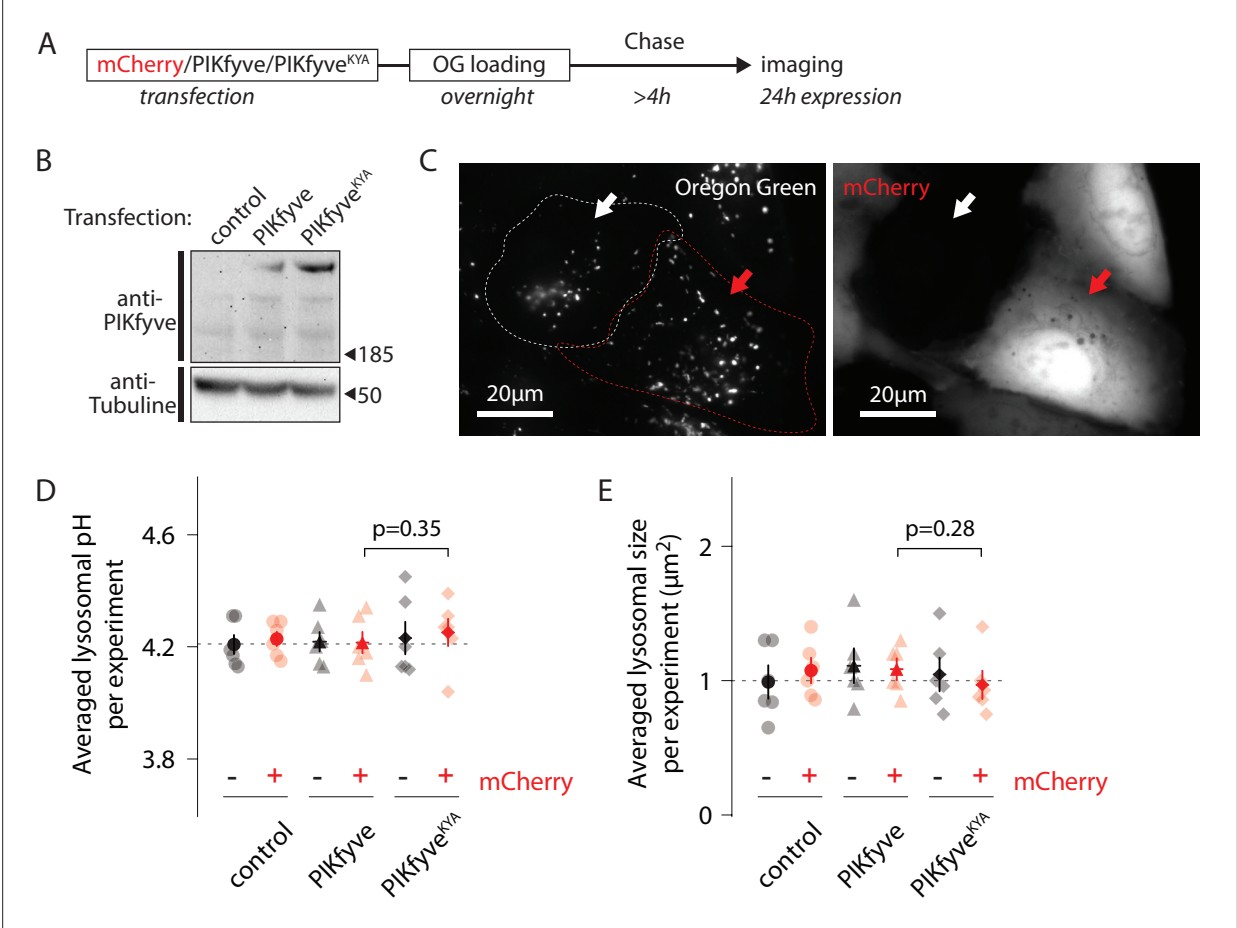

**Figure 4.** Increasing PI(3,5)P2 levels does not affect lysosomal pH. (**A**) Protocol timeline. U2OS cells were transfected with either mCherry alone, mCherry and PIKfyve, or mCherry and PIKfyve$^{KYA}$ and subsequently 'lysosome-loaded' with Oregon Green 488-dextran (OG). Cells were imaged after 24 hr expression to quantify lysosomal pH and size. (**B**) Western blot indicating PIKfyve or PIKfyve$^{KYA}$ expression level for each condition after 24 hr expression. Note that PIKfyve endogenous level (control) was too low to be detected. (**C**) Representative cells imaged by 445 nm laser excitation (OG channel, left image) and 640 nm laser excitation (mCherry channel, right image). The red arrow indicates one cell containing lysosomes loaded with OG (bright dots in OG channel) and expressing mCherry (bright cell in mCherry channel). The white arrow indicates one cell from the same batch containing lysosomes filled with OG but not expressing mCherry. Dotted lines delineate cell shape. (**D, E**) Comparison of lysosomal pH (**D**) or size (**E**) in control (circle), PIKfyve-transfected (triangle), and PIKfyve$^{KYA}$-transfected (diamond) conditions. For each condition, cells were separated into two populations based on the presence or absence of mCherry. Each symbol represents the averaged lysosomal pH or size of one experiment (six experiments; 8–11 cells per condition per experiment). p-Values: paired $t$-test between PIKfyve-mCherry and PIKfyve$^{KYA}$-mCherry conditions.

The online version of this article includes the following source data for figure 4:

**Source data 1.** Raw blot tubulin: merged images combining a light image of the blot itself and the chemiluminescent image of the blot probed with antitubulin antibody.

**Source data 2.** Raw blot tubulin: merged images of the same blot as above, but probed with an anti-PIKfyve antibody.

**Source data 3.** Labeled gel: the above gel images with labels.

average of multiple independent experiments in *Figure 5I*; 4.11 µm$^2$ ± 0.37 versu 2.04 µm$^2$ ± 0.29 for CLC-7 WT and KO cells, respectively; p=0.0017).

Overall, these results indicate that ClC-7 is involved in two important mechanisms downstream of PIKfyve. First, it contributes to setting the lysosomal pH, at least under some circumstances. Second, ClC-7 participates in the regulation of lysosomal size and/or lysosomal trafficking through a yet unknown mechanism.

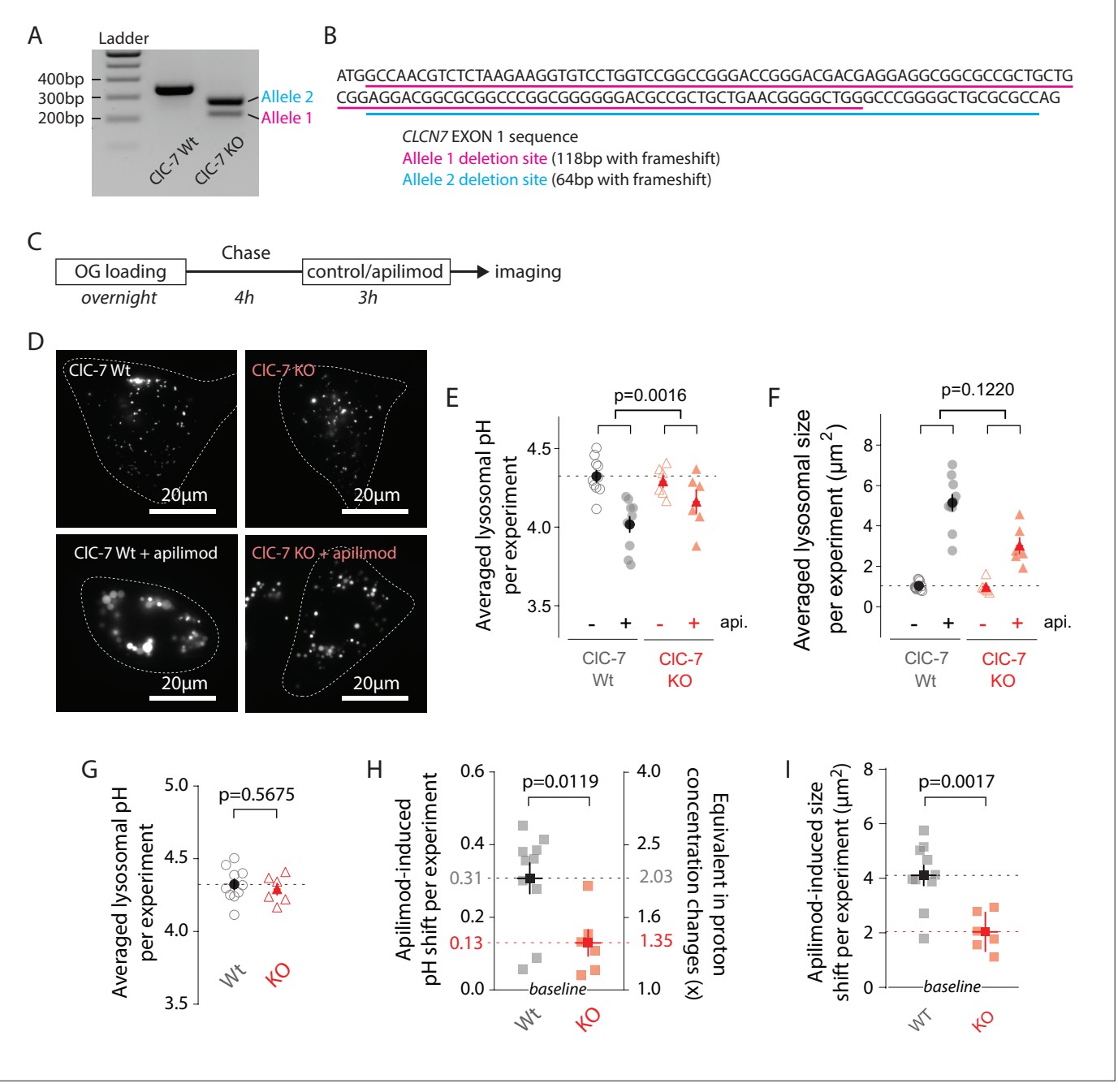

**Figure 5.** ClC-7 knockout (KO) cells display both lysosomal hyperacidification and swelling reduction. (**A**) Agarose gel electrophoresis of PCR product amplified from *ClCN7* deletion site of wildtype (ClC7 Wt, left) and *ClCN7* KO (ClC-7 KO, right) U2OS cells. (**B**) EXON1 sequence from *CLCN7* WT (black) and KO (purple and teal) alleles. (**C**) Protocol timeline. Cells were 'lysosome-loaded' with Oregon Green 488-dextran (OG) and treated for 3 hr with apilimod (100 nM, red) or its vehicle (0.25% DMSO, control) before imaging. (**D**) Images from a representative experiment: ClC-7 WT (left) or ClC-7 KO (right) cells acquired by 445 nm laser excitation. Bright objects represent OG-positive lysosomes in control (top) versus apilimod (bottom) conditions. Dotted lines delineate cell outlines. (**E, F**) Lysosomal pH (**E**) or size (**F**) from ClC-7 WT (gray circle) or ClC-7 KO (red triangle) cells in apilimod (filled symbols) versus control (empty symbols) in a representative experiment. Dark symbols are averages over all cells; each pale symbol represents the average lysosomal pH from one cell. p-Values for apilimod effects are obtained from two-way ANOVA. (**G**) There is no significant difference in pH between untreated WT and untreated KO cells (p=0.5675, unpaired *t*-test). For (**G - I**), dark symbols are averages over all experiments; each pale symbol represents the averaged lysosomal pH or size from one experiment (10 and 6 independent experiments for WT and KO conditions, respectively; each experiment represents 8–18 cells per condition). (**H, I**) Comparison of lysosomal pH shift (**H**) or size shift (**I**) induced by apilimod treatment in ClC-7 Wt (gray) versus ClC-7 KO (red) cells. Proton concentration change (**H**), [H$^+$]$_{change}$ was calculated from apilimod-induced pH shift ($\Delta$pH) using the following relation: $[H^+]_{change} = 10^{-\Delta pH}$. p-Values from unpaired *t*-test.

*Figure 5 continued on next page*

*Figure 5 continued*

The online version of this article includes the following source data and figure supplement(s) for figure 5:

**Source data 1.** Raw gel: agarose gel of PCR product amplified from CLCN7 deletion site of wildtype (WT) and CLCN7 knockout (KO) U2OS cells.

**Source data 2.** Labeled gel: same gel as above with labels.

**Figure supplement 1.** Validation of the U2OS ClC-7 knockout (KO) clone.

**Figure supplement 1—source data 1.** Raw gel: agarose gel of RT-PCR products of GAPDH (positive control), ClC-7 wildtype (WT), and ClC-7 knockout (KO) U2OS cells.

**Figure supplement 1—source data 2.** Labeled gel: labeled gel of above.

**Figure supplement 2.** .CLCN7 deletion does not affect 'lysosome-loading' with Oregon Green 488-dextran (OG) and does not alter pH measurements.

## PI(3,5)P2 inhibits ClC-7 transporter currents

In the simplest model for our observations, the effect of PI(3,5)P2 on ClC-7 would be via a direct interaction with the protein, thereby modulating its activity and its contribution to lysosomal acidification. Our results so far are consistent with a hypothesis in which PI(3,5)P2 inhibits ClC-7 transport activity: relief of this inhibition, as by PI(3,5)P2 removal, would activate ClC-7, allow excess Cl⁻ to enter the lysosome, and thereby provide excess counterions to facilitate further (and excessive) acidification by the V-type ATPase. A recently published cryo-EM structure of the human ClC-7/Ostm1 complex demonstrates that a direct interaction is indeed possible, revealing electron density compatible with a phosphoinositide bound to the protein (*Figure 6A and B*; *Schrecker et al., 2020*). With the headgroup located at the interface between the cytosolic and transmembrane domains (*Schrecker et al., 2020*), the positioning of the bound lipid is intriguing, especially given that a large body of evidence indicates that CLC cytoplasmic domains play important roles in the slow voltage-dependent gating observed in these proteins (*Bykova et al., 2006*; *Ludwig et al., 2013*). In this context, we next sought to directly test for a *functional* effect of PI(3,5)P2 on ClC-7 activity by measuring transport currents using patch-clamp electrophysiology. We transfected HEK-293 cells with a ClC-7 construct containing mutations to the N-terminal lysosome-targeting motif that enable functional transporters to be trafficked to the plasma membrane (ClC-7PM), allowing measurement of whole-cell currents using patch-clamp (*Leisle et al., 2011*; *Stauber and Jentsch, 2010*). In lysosomes, PI(3,5)P2 is found in the cytosolic leaflet of the membrane; thus, to introduce the phospholipid, we added 50 µM short-chain PI(3,5)P2 to the pipette solution (*Collins and Gordon, 2013*). We expected that after initial break-in the lipid would gradually diffuse into the cell and partition into the membrane over the span of a few minutes.

In the absence of PI(3,5)P2, repeated pulse families demonstrate the stability of the currents, with nearly 100% of the initial current remaining after 4 min (*Figure 6—figure supplement 1*). In contrast, with the signaling lipid in the pipette we observe substantial inhibition that develops over the first several minutes after break-in and reaches steady state by approximately 3 min, our chosen time point for measurement (*Figure 6—figure supplement 1*). Current families recorded in the absence (30 s after break-in) and presence (3 min after break-in) of PI(3,5)P2 reflect the effect of the signaling lipid on the transporter current. Recordings made without PI(3,5)P2 in the pipette showed little change between the two time points; activation and deactivation kinetics and current magnitudes remained mostly constant, again demonstrating that the ClC-7 currents are stable over time (*Figure 6E*). However, including 50 µM PI(3,5)P2 in the pipette resulted in an ~39% decrease in maximum voltage-activated currents and a 66% decrease in the amplitude of tail currents following repolarization (*Figure 6C and E*). The deactivation rate was also significantly faster in the presence of PI(3,5)P2 (*Figure 6—figure supplement 2A*). The effects of PI(3,5)P2 on maximal current and tail current amplitude were dose-dependent with increasing inhibition by PI(3,5)P2 over concentrations between 25 and 100 µM (*Figure 6—figure supplement 2E*). PI(3,5)P2 inhibition of ClC-7 appears to be specific: neither PI3P nor PI(4,5)P2 yielded detectable inhibition of the ClC-7 currents (*Figure 6G and H*, *Figure 6—figure supplement 2C and D*). Because the lipid bound to ClC-7 in the structure was modeled as a PI3P (due to steric clashes between the 5-phosphate and adjacent residues when PI(3,5)P2 was modeled), its lack of effect on ClC-7 currents was surprising. Since the lipid in the cryo-EM structure was not added during the purification, it must have been carried through the preparation bound to ClC-7 and should be tightly bound, but the lack of effect in our experiments suggests, at best, a low affinity. To further test for an interaction with PI3P, we therefore attempted competition experiments with equimolar mixes of PI(3,5)2 and PI3P; the presence of PI3P in these experiments did not detectably affect the

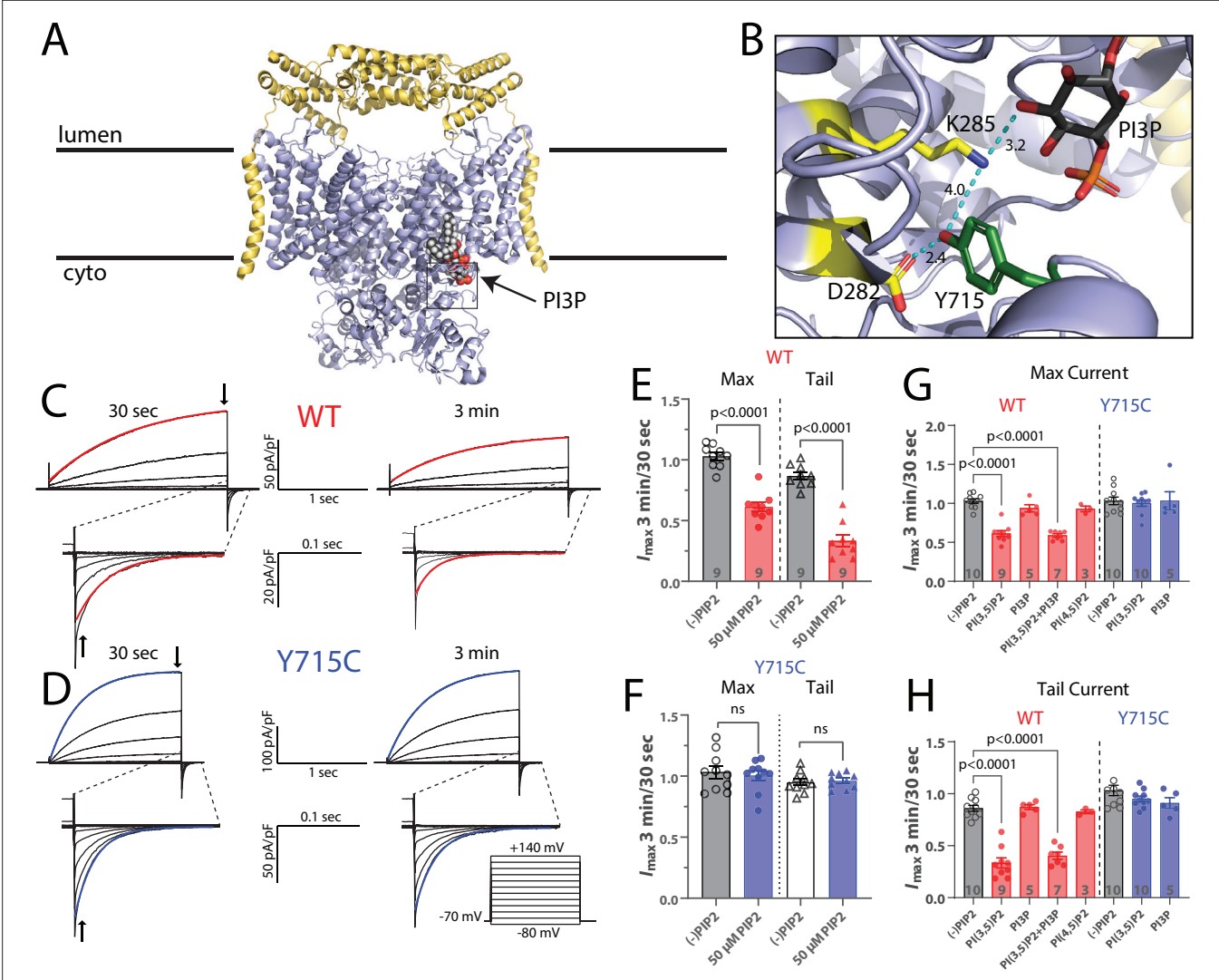

**Figure 6.** PI(3,5)P2 inhibits ClC-7-mediated currents. (**A**) Cryo-EM structure of human ClC-7 (pale blue) and Ostm1 (yellow) (PDB 7JM7). PI3P is modeled bound at the interface between the transmembrane domain and the cytosolic CBS domains. (**B**) Enlarged view of the PI3P binding region boxed in (**A**). The structure reveals a hydrogen-bonding network involving the phospholipid headgroup and adjacent residues, including Y715. Measurement distances in angstroms are indicated. (**C, D**) Representative whole-cell current traces recorded from HEK-293 cells transfected with (**C**) wild-type or (**D**) Y715C mutant ClC-7. After allowing the pipette solution containing 50 μM PI(3,5)P2 to diffuse into the cell for 3 min, maximum voltage-activated current magnitude and tail current magnitude on returning to holding potential were reduced in currents from wildtype but not mutant ClC-7. Red or blue lines indicate a single exponential fit to obtain time constants. (**E, F**) The ratio of current magnitudes measured at 3 min and 30 s after cell break-in from (**E**) wildtype and (**F**) Y715C-transfected cells as indicated by arrows in (**C**) and (**D**). Each point represents the ratio of two successive recordings from a single cell. Error bars represent standard error of the mean, and p-values are calculated from a two-tailed unpaired Student's t-test. (**G, H**) Effect of 50 μM phospholipids on the change in (**G**) maximum current and (**H**) tail current after a +140 mV pulse as measured by the ratio of the indicated values 3 min and 30 s after cell break-in. Error bars represent standard error of the mean, and adjusted p-values are calculated by Tukey's method.

The online version of this article includes the following figure supplement(s) for figure 6:

**Figure supplement 1.** Time course of PI(3,5)P2 inhibition.

**Figure supplement 2.** Effect of phosphoinositides on ClC-7 currents in HEK cells.

**Figure supplement 3.** Mouse TPC1 currents evoked by PI(3,5)P2.

inhibition by PI(3,5)P2, suggesting a specific effect of the doubly phosphorylated lipid (*Figure 6G and H*, *Figure 6—figure supplement 2C and D*). We conclude that PI(3,5)P2 specifically inhibits ClC-7 transport currents.

The concentrations of PI(3,5)P2 required for effective inhibition in our experiments are relatively high, with an EC50 of approximately 30 μM. We sought to distinguish the possibility of low-affinity interaction with ClC-7 from the alternative explanation that the concentration of lipid in the patch pipette does not deliver a matching level to the plasma membrane by calibrating our system against a protein with a known dose response to PI(3,5)P2, in this case the lysosomal cation channel TPC1 (*She et al., 2018*). PI(3,5)P2 activates currents of a mutant TPC1 expressed in HEK cells when added to the pipette in conditions similar to those we use. In published work, the EC50 of this activation is approximately 145 nM. In our hands, PI(3,5)P2 activates TPC1 at higher pipette concentrations, with an EC50 of 7–10 μM (*Figure 6—figure supplement 3*), suggesting less efficient delivery of the lipid to the plasma membrane in our system. In this setting, our measured EC50, though higher than that of TPC1, is in a similar range. In either case, we expect that the final concentration of PI(3,5)P2 in the plasma membrane is substantially, likely more than an order of magnitude, lower than that in the pipette solution, leaving the actual binding affinity uncertain.

## The pathogenic mutation Y715C abolishes PI(3,5)P2 inhibition of ClC-7

Recently, a novel disease-causing mutation in ClC-7, Y715C, was reported in two patients (*Nicoli et al., 2019*). At the lysosomal level, this variant is characterized by enlarged vacuoles and hyperacidified lysosomes, a phenotype remarkably similar to that observed in apilimod-treated cells. Consistent with these features, currents measured from cells expressing ClC-7-Y715C were larger than those expressing wildtype ClC-7, suggesting that enhanced chloride/proton exchange could allow more Cl⁻ to enter the lysosome and facilitate excess acidification, thereby contributing to the observed lower pH levels (*Nicoli et al., 2019*). Examination of the human ClC-7 structure reveals that the side chain of tyrosine 715, located on the cytoplasmic domain, is within 10 Å of the inositol ring of the bound PIP, and that it is involved in a hydrogen-bonding network with ring hydroxyls and several TM-domain residues (*Figure 6B*).

We reasoned that this network might play an important role in coupling the binding of PI(3,5)P2 to changes in transport currents. If so, the Y715C mutation might perturb this interaction and limit the inhibitory effect of PI(3,5)P2, resulting in the more active phenotype we previously described (*Nicoli et al., 2019*). To test this possibility, we repeated the PIP2 patch-clamp electrophysiology experiments described above after introducing the Y715C mutation. For currents from ClC-7PM Y715C, the inhibitory effect of PI(3,5)P2 we observed for WT transporters is no longer detectable; the phosphoinositide had essentially no effect on the mutant ClC-7 currents under these conditions (*Figure 6D and F*, *Figure 6—figure supplement 2D*). This result suggests that the Y715C mutation either weakens the binding of PI(3,5)P2 to ClC-7 or disrupts the coupling between lipid binding and the transporter gating. As with wildtype ClC-7, PI3P also had no detectable effect on Y715C mutant currents (*Figure 6G and H*, *Figure 6—figure supplement 2C and D*). Even increasing the pipette concentration of PI(3,5)P2 to 100 μM did not affect Y715C currents (not shown). These results demonstrate a mechanistic connection between the lysosomal effects observed under PIKfyve inhibition and those observed in patients carrying the Y715C mutation: in the former case, a decrease in available PI(3,5)P2 relieves a tonic inhibition of ClC-7 by the lipid, and in the latter, PI(3,5)P2 may be present but it is unable to elicit its usual inhibition of transport.

Close examination of the human ClC-7 structure reveals that Y715 is engaged in a network of H-bonds with residues on the protein's transmembrane domain that extends to the 5′ hydroxyl on the PIP inositol ring (*Figure 6B*). We tested the importance of the hydroxyl group and phenyl ring on this tyrosine by mutating it to a phenylalanine or to a serine, respectively, and measured transporter currents using the ClC-7PM expressed in *Xenopus* oocytes with two-electrode voltage clamp (*Figure 7*). Removing either the hydroxyl group (Y715F) or the phenyl ring (Y715S) both increased maximal current at +80 mV and sped current activation. For each mutation we examined, the transporter current had characteristics nearly identical to that of the original Y715C mutant, suggesting that the WT tyrosine is essential at this position. Indeed, multiple other mutations at this position all resulted in currents very similar to that of Y715C, strongly supporting the importance of the WT tyrosine for proper transporter function (*Figure 7*, *Figure 7—figure supplement 1*).

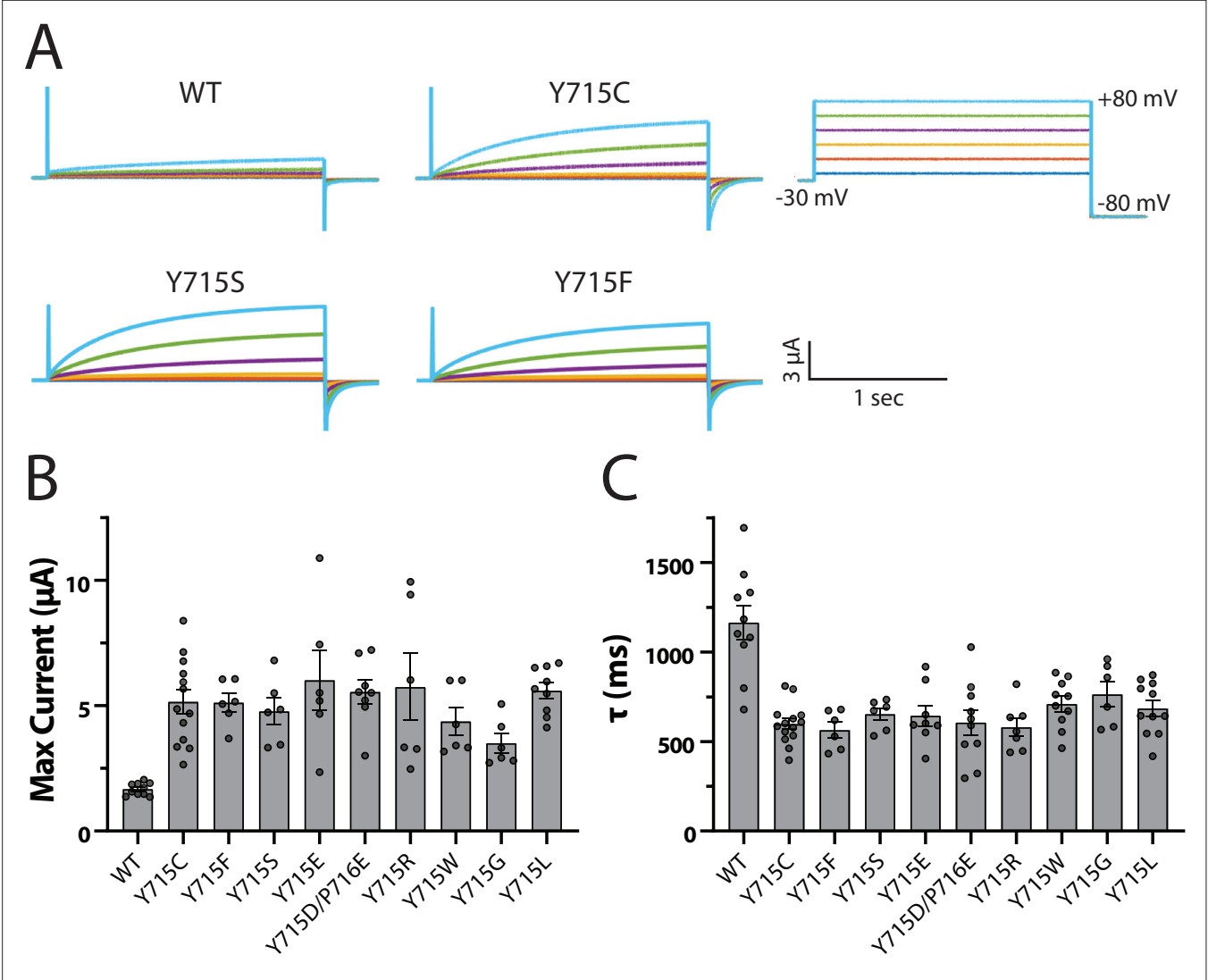

**Figure 7.** Y715 is required for proper ClC-7 function. Y715x mutations were introduced to the plasma membrane-targeted ClC-7 and currents measured in *Xenopus* oocytes under two-electrode voltage clamp. (**A**) Representative voltage families recorded from oocytes injected with the indicated construct. (**B**) Maximum current measured from ClC-7 expressing oocytes at the end of a 2 s +80 mV voltage pulse. (**C**) Activation time constant at +80 mV obtained from a single exponential fit to the current trace. Each dot represents a value measured from an individual oocyte; 2–5 independent batches of oocytes were recorded for each mutant.

The online version of this article includes the following figure supplement(s) for figure 7:

**Figure supplement 1.** The side chain of Y715 is essential for preserving wildtype function of ClC-7.

## Discussion

In this study, we demonstrated that acute pharmacological inhibition of PIKfyve and consequent depletion of PI(3,5)P2 induces a lysosomal hyperacidification of 0.3–0.5 pH units, and that the effect is largely due to contributions from ClC-7. Our mechanistic experiments revealing direct inhibition of ClC-7 by PI(3,5)P2 suggest that tonic PI(3,5)P2 inhibition of the ClC-7 is relieved by the lipid depletion caused by apilimod inhibition of PIKfyve. We observed hyperacidification in multiple cell types, including U20S osteosarcoma cells, human neonatal fibroblasts, and SUDHL B-cell non-Hodgkin's lymphoma cells. Our results clearly establish that PIKfyve inhibition can affect lysosomal pH. There are several possible reasons why previous efforts to identify such changes were unsuccessful. First, it is essential to use a dye with sensitivity at the low end of the lysosomal pH range. *Ho et al., 2015* used FITC as their pH-sensing dye; with a pKa around 6.0 the dye is minimally sensitive to pH below about 4.5 so changes in the range we observe would be difficult to detect. Recently, Sharma et al.

measured lysosomal pH with OG and concluded that apilimod or a new PIKfyve inhibitor named WX8 does not affect lysosomal pH of RAW or U2OS cells (*Sharma et al., 2019*). However, their experiments yield untreated lysosomal pH values of 5.25 ± 0.05 units in both cell lines; this is above the range we observe and may indicate a problem with calibration or image processing. Nonetheless, that group did observe a (significant) hyperacidification of ~0.2 pH units with one inhibitor (WX8), consistent with our findings. Additional variability may result from cell-type differences in acidification control or in the role of PIKfyve (*Sbrissa and Shisheva, 2005*).

Apilimod-induced vacuoles originate from endolysosomes and lysosomes through alterations in the tubulation/fission processes that constantly reform these organelles (*Bissig et al., 2017*; *Choy et al., 2018*; *Sharma et al., 2019*). Although the molecular mechanism for these processes remains unknown, Compton et al. showed (and our results replicated) that the proton pump V-ATPase must be active to produce vacuoles (*Compton et al., 2016*), though it has been suggested that these effects could result from off-target effects of BafA1 (*Mauvezin et al., 2015*). Our results using pH-modifying agents early in vacuole development clearly demonstrate that the hyperacidic luminal pH resulting from apilimod treatment is neither responsible for initiation of vacuole formation, nor for the increase in vacuole size, as alkalinizing lysosomes to pH values above about pH 5.5 had no impact on these effects. In further support of a bifurcation of these processes, lysosomal tubulation and fission events (size decrease) precede the return to normal pH. However, lysosomal size and pH are not completely decoupled, as indicated by the observation that ClC-7 knockout dampens both apilimod-induced size and pH changes, as well as by the effects of the disease-causing mutation ClC-7 Y715C (*Nicoli et al., 2019*). This shift could reflect alteration of lysosomal membrane electrical potential or changes of luminal ion concentrations, in turn affecting the trafficking and/or inducing osmotic stress (*Di Zanni et al., 2021*; *Freeman et al., 2020*; *Sartelet et al., 2014*).

Our electrophysiology experiments reveal a direct and specific inhibitory effect of PI(3,5)P2 on ClC-7 activity. Based on the changes caused by PI(3,5)P2 to the kinetics of slow ClC-7 voltage-dependent activation and deactivation, we suggest that the lipid tunes this slow gating process rather than affecting the transport cycle itself. Notably, in the human ClC-7 structure the bound lipid is modeled as a PI3P, a lipid that did not perceptibly inhibit ClC-7 transport in our experiments and did not compete with PI(3,5)P2 at concentrations we could achieve. Further work will be necessary to determine the role, if any, of PI3P in ClC-7 function. The headgroup of the bound lipid in the structure participates in a hydrogen bonding network that encompasses residues both on the TM domain and the cytoplasmic domain of the protein, and which includes the Y715 residue that is mutated in the gain-of-function patient. Our observations here that the Y715C mutation abolishes PI(3,5)P2 inhibition and that the Y715 position is exquisitely sensitive to mutation suggest a model wherein ClC-7 voltage-dependent gating comprises substantial conformational changes involving this interface, changes that we infer are modulated by the presence and identity of the bound lipid. Perhaps upon binding PI(3,5)P2 the 5-phosphate revises the H-bond network and induces a gating-controlling conformational change to inhibit the transporter. This model is consistent with a large body of evidence for a role of the cytoplasmic domain in slow gating of ClC-7, including a report that the ClC-7 C-terminal domain is necessary for voltage-dependent gating that corroborates the importance of this interface (*Di Zanni et al., 2021*; *Ludwig et al., 2013*; *Sartelet et al., 2014*).

The work we present here advances our understanding of the role of ClC-7 in lysosomal acidification. Previous work has suggested that ClC-7 forms part of the lysosomal counterion pathway based on increases in lysosomal pH in cells with ClC-7 knocked down using siRNA (*Graves et al., 2008*), as well as the effect of Cl⁻ removal from the bathing media in isolated lysosomes (*Ohkuma et al., 1982*). However, cells from multiple tissues in ClC-7 KO mice show normal lysosomal pH (*Kasper et al., 2005*; *Wartosch et al., 2009*), a conclusion reinforced by the retention of normal pH in the untreated ClC-7 KO U2OS cells we examined here. Our results suggest a possible explanation for this discrepancy: we hypothesize that ClC-7 is largely inhibited by PI(3,5)P2 in normal cellular conditions, preventing it from strongly influencing baseline lysosomal pH, but is activated by stimuli that decrease PI(3,5)P2 concentrations, thereby providing additional Cl⁻ as a counterion and facilitating further lysosomal acidification. This model is supported by the hyperacidification in patient-cell lysosomes carrying the Y715C gain-of-function mutation (*Nicoli et al., 2019*). Such a mechanism could complement that observed in microglia, where trafficking of ClC-7 to the lysosome upon activation by inflammatory stimuli caused acidification (*Majumdar et al., 2011*). Of

course, an alternate explanation is that ClC-7 is indeed the central locus of the counterion pathway, but compensatory changes in expression or regulation of other channels/transporters in the ClC-7 KO account for the normal lysosomal pH in those cells and animals. The nature of the physiological stimuli for changes in PI(3,5)P2 to regulate lysosomal pH remains to be determined, though the PIKfyve kinase has been shown to participate in multiple metabolic pathways (*McCartney et al., 2014a*).

A complementary possibility is that the effects of ClC-7 activity on lysosomal [Cl⁻] are important in some or all of the effects we observe. Due to the lack of a lysosomal pH change in the ClC-7 KO mouse, as well as the significant defects in mice carrying an uncoupled ClC-7, Jentsch and colleagues have proposed that ClC-7's primary function could be to regulate luminal [Cl⁻] (*Weinert et al., 2010*; *Weinert et al., 2014*). We have not yet been successful in measuring lysosomal Cl⁻ in our system; we leave open the possibility that increased flux through ClC-7 could lead to increased luminal osmotic strength, which, in turn, contributes to vacuole growth. However, the dissociation between vacuole size and hyperacidification and the continued size increase in the ClC-7 KO cells perhaps argues against this hypothesis.

Regulation of lysosomal pH is critical for normal cellular function; the complexity of this regulation is fertile ground for further study. Though the proton-pumping V-Type ATPase is clearly the metabolic driver of acidification, other factors, including the counterion pathway, are also critical. In recent years, a host of candidates for the counterion pathway have been identified, including ion channels like TRPML1 (*Dong et al., 2010*), TPC1 and TPC2 (*Wang et al., 2012*), and TMEM175 (*Cang et al., 2015*), and ion transporters like ClC-7 (*Graves et al., 2008*); indeed, several of these channels are also known to be modulated by PI(3,5)P2 (*Fine et al., 2018*; *She et al., 2018*; *Wang et al., 2012*). However, their physiological functions and contributions to different lysosomal processes are still emerging. Interestingly, knockout of FIG4, the phosphatase that degrades PI(3,5)P2, leads to a phenotype that shares many characteristics with the gain-of-function ClC-7 Y715C (*Chow et al., 2007*; *Lenk et al., 2019*). Paradoxically, the FIG4 KO demonstrates *reduced* levels of PI(3,5)P2; thus, its phenotype may also reflect increased activity of ClC-7 due to decreased inhibition by the signaling lipid.

Besides pH maintenance, lysosomal ions clearly contribute to multiple other processes, including triggering membrane fusion (*Dong et al., 2010*), driving the export of lysosomal degradation products (*Sagné and Gasnier, 2008*), maintaining osmotic balance (*Freeman et al., 2020*), and mediating lysosomal excitability (*Cang et al., 2014*), suggesting a need for coordinated control of multiple pathways. The work we present here uncovers a wealth of possibilities to study fundamental processes underlying lysosomal pH control and hints at complex interactions between multiple ion pathways to balance competing physiological requirements. Much work remains to understand the interplay of these factors and their pleiotropic effects.

## Materials and methods
### Chemicals
Apilimod (MedChemExpress); WX8 (gift from Juan S. Bonifacino lab) Oregon Green 488-dextran 10,000 mW (OG; Thermo Fisher); chloroquine, BafA1, monensin, and nigericin (Sigma-Aldrich); LysoTracker Blue DND-22 (Invitrogen); Magic Red cathepsin B essay (ImmunoChemistry Technologies); PI(3,5)P2 diC8, PI(4,5)P2 diC8, PI3P diC8 (Echelon Biosciences).

### Cell culture
U2OS cells (HTB-96) were purchased from ATCC and cultured in McCoy's 5A Modified Medium supplemented with 10% fetal bovine serum (FBS) and 100 U/mL penicillin-streptomycin (all from Gibco) at 37°C in 5% $CO_2$ and 10% humidity. Cells were passed two times per week using TrypLE (Gibco) and were used for experiments with a passage number lower than 40. Human primary neonatal fibroblasts were also purchased from ATCC (PCS-201-010) and cultured in DMEM/F12 GlutaMAX medium supplemented with 10% FBS and 100 U/mL penicillin-streptomycin at 37°C in 5% $CO_2$ and 10% humidity. Cells were passaged twice weekly; cells used in three independent apilimod treatment experiments were at passages 10, 12, and 16. All culture reagents were obtained from Gibco.

## OG lysosome-loading procedure

The day before imaging, cells were incubated overnight in a Nunc Lab-Tek II 8-well chamber (Thermo Fisher) in culture medium supplemented with 1 mg/mL OG for 15–16 hr. OG was subsequently chased by washing for at least 4 hr before starting a protocol. For fibroblast experiments, cells were incubated with 2 mg/mL OG.

## Cell imaging

Live-cell imaging was performed at room temperature using a Nikon eclipse-Ti microscope with a ×60 objective (Apo TIRF, Nikon) and a digital camera (ORCA-flash 4.0, Hamamatsu). Cells were imaged in an imaging media containing (in mM) 150 NaCl, 5 KCl, 1 CaCl$_2$, 1 MgCl$_2$, 20 HEPES, 10 glucose, and 30 sucrose, adjusted pH 7.2–7.4 with NaOH. For pH measurements, images were acquired in epifluorescent mode using a 445 nm laser (OBIS, Coherent), a 488 nm laser (Sapphire, Coherent), and an >500 nm emission filter (ET500lp, Chroma). For each cell, separate images with each laser were obtained within 1 s of each other. For experiments using mCherry, we used the same excitation filterset to perform pH measurements but used a 535 ± 35 nm emission filter (ET535/70m, Chroma) to eliminate crosstalk from the red channel. A SOLA light engine (Lumencor) was used with an mCherry cube (49008, Chroma) to image for mCherry fluorescence. Image acquisition was performed using NIS-Element software (Nikon). For fibroblast lysosome pH measurements, epifluorescence images were acquired using a CoolLED pE-4000 LED system. Images were acquired by excitation using the 435 nm LED or 490 nm LED with a 488/10 nm BrightLine single-band bandpass excitation filter (Semrock) and the 435 nm LED.

## Image analysis

Intensity and size of cell's OG-positive objects were extracted using the software Slicer2D developed by Calvin Johnson and Huey Cheung from the Center for Information Technology at the National Institutes of Health (*Lau et al., 2013*). Briefly, local background of 445 nm and 488 nm images were background-subtracted and subsequently threshold using the Otsu method (*Otsu, 1979*) to select the region of interest (ROI) corresponding to individual lysosomes. Each ROI is paired between the two images and a filter comparing paired ROI centroid and intensity is applied to discard single-image ROI or ROI containing low-intensity pixels value. Size, intensity, and 488/445 ratio of each ROI were then calculated.

## pH calibration curve

Just after imaging cells in protocol-based conditions, imaging media were washed out and replaced sequentially by a set of pH calibrated buffers supplemented with monensin (7.0 µg/mL, dissolved in ethanol) and nigericin (7.5 µg/mL, dissolved in ethanol; 0.29% ethanol in final solution) to equilibrate pH across cell and organelle membranes. pH calibration buffers composition: pH 7.0: imaging media (IM) plus 40 mM dibasic sodium phosphate (NaP) and 10 mM Tris-maleate (TM); pH 6.0: IM plus 25 mM NaP and 25 mM TM; pH 5.0: IM plus 5 mM NaP and 45 mM TM; pH 4.5: IM plus 30 mM NaP and 20 mM citrate; pH 4.0: IM plus 26 mM NaP and 24 mM citrate; pH 3.0: IM plus 20 mM NaP and 30 mM citrate. For each buffer, lysosomal pH measures were obtained on 4–5 different cells. 488/445 nm ratios were then plotted in function of pH buffers and fitted by a four-parameter sigmoidal function curve using Sigma plot v12 (Systat Software Inc).

## ClC-7 KO CRISPR clone engineering

To knock out ClC-7 at the genomic level in U2OS cells, two guide RNA plasmid constructs were created using lentiCRISPRv2 (*Shalem et al., 2014*; *Shalem et al., 2014*). Target sites for deletion in EXON1 of *CLCN7* were designed using a CRISPR design website (http://crispr.mit.edu/). The resulting target sequences were as follows: 5'-TGTCCTGGTCCGGCCGGGACCGG-3'; 5'-ACGCCGCT GCTGAACGGGGCTGG-3'. Guide oligonucleotide sequences were: T-1F, 5'-CACCGTGTCCTGGTCC GGCCGGGAC-3'; T-1R, 5'-AAACGTCCCGGCCGGACCAGGACAC-3'; T-2F, 5'-CACCGACGCCGC TGCTGAACGGGGC-3'; T-2R, 5'-AAACGCCCCGTTCAGCAGCGGCGTC-3'. As described in Zhang lab protocol (*Shalem et al., 2014*), each guide oligo pairs were then annealed, cloned, and transformed to Stbl3 bacteria. The positive constructs were identified and confirmed by sequencing. U2OS

cells transfection with engineered lentiCRISPRv2 were done using FuGENE 6 transfection reagent (Promega). Positive clones were selected using puromycin and subsequently isolated.

## U2OS cell genotyping and RT-qPCR amplification

Genotyping of ClC-7 KO U2OS clone was performed using an Extract PCR-Kit (Bioline) with the following primers: GAGAACAAACACGGGGGCA; GCGTTCCCGAGTCCACC. Isolated total RNA from WT and KO U2OS cells (QIAGEN RNeasy Mini Kit) reverse-transcribed into cDNA (BIORAD iScript cDNA Synthesis Kit) and primers (indicated as followed) were used to quantify mRNA expression levels by real-time PCR using iQ SYBR Green Supermix (Bio-Rad) and CFX96 Touch Real-Time PCR Detection System (Bio-Rad). Primers used: GAPDH: 5'-GTCTCCTCTGACTTCAACAGCG-3'; 5'-ACCACCCTGTTGCTGTAGCCAA-3'. ClC-7: 5'-CCCACTCCAGCTCTTCTGTG-3'; 5'-ATAGGAGCCTGGTGGGTCATG-3'. Relative expression of ClC-7 mRNA was calculated by normalizing the expression values of WT and KO to those of GAPDH.

## mCherry, PIKfyve constructs, and transfection procedure

PIKfyve (sequence id: AY457063.1) and PIKfyve-KYA hyperactive mutant (E1620>K, N1630>Y, S2068>A) were cloned into PCMV-HA-N vector (Clontech) and gifted by Lois Weisman lab. mCherry (sequence id: AIL28759.1) was cloned into PCAGEN vector (Addgene 11160) and gifted by Kenton Swartz lab. U2OS cells were transfected in DMEM media with Lipofectamine 2000 (ThermoFisher).

## PIKfyve Western blot

Approximately $2 \times 10^6$ U2OS cells expressing or not PIKfyve or PIKfyve$^{KYA}$ were collected and sonicated to extract total proteins. 100 µg of total proteins was then subjected to electrophoresis in 4–12% Bis-Tris Plus Gels (Novex, NW00125BOX) and transferred onto PVDF membrane using a Trans-Blot Turbo Transfer System (Bio-Rad). After blocking with 5% non-fat milk, PVDF membrane was incubated at 4°C with 1:1000 ratio of PIKFYVE antibody (Invitrogen, PA5-13977) or 1:1000 ratio of α/β-tubulin antibody (Cell Signaling, 2148). Both primary antibodies were probed using goat anti-rabbit IgG secondary antibody conjugated with horseradish peroxidase (Jackson ImmunoResearch, West Grove, PA) and visualized by chemiluminescence (SuperSignal West Pico Chemiluminescent Substrate, Thermo Fisher). Images were obtained using an Odyssey FC Imager (Dual-Mode Imaging System; 2 min integration time).

## Electrophysiology

HEK-293 (ATCC CRL-1573) cells were cultured in growth medium consisting of 90% Dulbecco's modified Eagle's medium, 10% FBS, 100 units mL$^{-1}$ of penicillin-streptomycin, 4 mM L-glutamine, and 1 mM sodium pyruvate (Gibco). Cells were cultured in 35 mm polystyrene dishes (Falcon) at 37°C in the presence of 5% $CO_2$. Cells were transiently co-transfected with 700 ng each of human wildtype ClC-7 or ClC-7-Y715C in a pIRES2–EGFP vector and human Ostm1 in a pCMV-6 vector. Both ClC-7 constructs also included mutations to two lysosome-targeting motifs in the N terminus to cause trafficking to the plasma membrane (*Leisle et al., 2011*; *Stauber and Jentsch, 2010*), allowing for whole-cell measurements. Transfection was accomplished using Lipofectamine LTX with Plus reagent (Invitrogen). Transfection was performed approximately 40 hr prior to experiments.

On the day of patch-clamp experiments, cells were released from culture dishes by brief exposure to 0.25% trypsin/EDTA, resuspended in supplemented DMEM, plated on glass coverslips, and allowed to recover for 1–2 hr at 37°C in 5% $CO_2$. Whole-cell voltage-clamp current measurements were performed using an Axopatch 200B amplifier (Axon Instruments) and pClamp 11.1 software (Axon Instruments). Data were acquired at 10 kHz and filtered at 1 kHz. Patch pipettes were pulled using a P-97 laser puller (Sutter Instruments) from borosilicate glass capillaries (World Precision Instruments) and heat-polished using an MF-200 microforge (World Precision Instruments). Pipette resistance was 2–5 MΩ in the extracellular solution. A reference electrode was placed in a separate chamber containing extracellular solution and connected to a 2% agar bridge made from extracellular solution. Extracellular solution consisted of (in mM) 130 NaCl , 5 KCl, 1 MgCl$_2$, 1 CaCl$_2$, 20 HEPES, and 20 glucose, with the pH adjusted to 7.4 using NaOH and the osmolality adjusted to 310 mOsmol using sucrose. Pipette solution contained (in mM) 110 CsCl, 10 NaCl, 2 MgCl$_2$, 1 EGTA, and 40 HEPES; pH was adjusted to 7.2 with CsOH, and osmolality was adjusted to 300 mOsmol using sucrose. Chemicals

were obtained from Sigma. Osmolality was measured using a Vapro 5600 vapor pressure osmometer (Wescor). For measurements in the presence of PIPs, PI(3,5)P2 diC8 (Echelon Biosciences P-3508), PI3P (Echelon Biosciences P-3008), or PI(4,5)P2 (Echelon Biosciences P-4508) was dissolved in water at a concentration of 2 mM and stored at –80°C in small aliquots; on the day of experiments, stock PIP2 was diluted into pipette solution to a final concentration of 50 µM and kept on ice during the course of the experiment.

Mouse TPC1 was obtained from Origene in a pCMV6 vector (MR210795) and subcloned into the pIRES2-EGFP vector. Mutations Leu11Ala and Ile12Ala have previously been shown to increase channel expression to the plasma membrane and were introduced by standard mutagenesis methods. 1.5 µg of DNA was transfected into HEK293 cells and electrophysiology experiments were carried out as above. For mTPC1 experiments, extracellular solution contained (in mM) 145 sodium methanesulfonate , 5 NaCl , 1 $MgCl_2$, 1 $CaCl_2$, and 10 HEPES, with the pH adjusted to 7.4 using NaOH and the osmolality adjusted to 310 mOsmol using sucrose. Pipette solution contained (in mM) 145 sodium methanesulfonate , 5 NaCl , 4 $MgCl_2$, 1 EGTA, and 10 HEPES; pH was adjusted to 7.4 with NaOH, and osmolality was adjusted to 300 mOsmol using sucrose.

## Statistical analysis

Data are reported as mean ± SEM and were analyzed using Prism 8 (GraphPad). Each statistical test used is indicated in the legend of the corresponding figure.

## Acknowledgements

We thank LS Weisman for providing us the PIKfyve[KYA] construct, K Swartz for providing the mCherry construct, and ML DePamphilis for the WX8 compound. K Swartz and C Lingle provided critical readings of the manuscript and valuable discussions. This work was supported by the NINDS intramural program.

## Additional information

### Funding

| Funder | Grant reference number | Author |
| --- | --- | --- |
| National Institute of Neurological Disorders and Stroke | | Xavier Leray<br>Jacob K Hilton<br>Kamsi Nwangwu<br>Alissa Becerril<br>Vedrana Mikusevic<br>Gabriel Fitzgerald<br>Anowarul Amin<br>Mary R Weston<br>Joseph A Mindell |

The funders had no role in study design, data collection and interpretation, or the decision to submit the work for publication.

### Author contributions

Xavier Leray, Conceptualization, Data curation, Formal analysis, Investigation, Methodology, Validation, Visualization, Writing – original draft, Writing – review and editing; Jacob K Hilton, Conceptualization, Formal analysis, Investigation, Methodology, Validation, Visualization, Writing – original draft, Writing – review and editing; Kamsi Nwangwu, Formal analysis, Investigation, Methodology, Visualization, Writing – review and editing; Alissa Becerril, Formal analysis, Investigation, Methodology, Validation, Writing – review and editing; Vedrana Mikusevic, Gabriel Fitzgerald, Investigation, Writing – review and editing; Anowarul Amin, Mary R Weston, Investigation, Methodology; Joseph A Mindell, Conceptualization, Formal analysis, Funding acquisition, Project administration, Resources, Supervision, Writing – original draft, Writing – review and editing

### Author ORCIDs

Xavier Leray ⓘ http://orcid.org/0000-0003-2107-6082

Jacob K Hilton (iD) http://orcid.org/0000-0003-1931-9516
Kamsi Nwangwu (iD) http://orcid.org/0000-0002-3446-8539
Vedrana Mikusevic (iD) http://orcid.org/0000-0002-9666-9571
Joseph A Mindell (iD) http://orcid.org/0000-0002-6952-8247

## Decision letter and Author response

Decision letter https://doi.org/10.7554/eLife.74136.sa1
Author response https://doi.org/10.7554/eLife.74136.sa2

## Additional files

### Supplementary files

• Transparent reporting form

### Data availability

All analyzed data are included in the manuscript.

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
