## [Editor Report]

This article will be of broad interest to readers in the field of lysosomal biology. It demonstrates that the lysosomal phosphoinositide PI(3,5)P2 tonically inhibits the CLC-7/Ostm1 chloride/proton antiporter. Relief of this inhibition leads to lysosome overacidification and enlargement and can explain the phenotype of a disease-causing ‘gain-of-function’ mutation in CLC-7. Together, these results suggest that the PI(3,5)P2/CLC-7 interaction may play a regulatory role in lysosome homeostasis.

---

## [Decision Letter]

**Decision letter after peer review:**

Thank you for submitting your article "Tonic inhibition of the chloride/proton antiporter ClC-7 by PI(3,5)P2 is crucial for lysosomal pH maintenance" for consideration by *eLife*. Your article has been reviewed by 3 peer reviewers, and the evaluation has been overseen by a Reviewing Editor and Richard Aldrich as the Senior Editor. The following individuals involved in review of your submission have agreed to reveal their identity: Haoxing Xu (Reviewer #2); Elena Oancea (Reviewer #3).

Essential revisions:

1) The concentrations of PI(3,5)P2 used in the patch clamp experiments are very large compared to what are thought to be physiological concentrations, based on studies investigating other PI(3,5)P2-sensitive channels and transporters. For comparison, reported values for activation of lysosomal cation channels for PI(3,5)P2 are three orders of magnitude smaller for mouse TRPML1 (Dong et al., JBC, 20210), and more than two orders of magnitude smaller for *Drosophila* TRPML1 and human TPC2 (Feng et al., JBC, 2013; Wang et al., Cell, 2012; Boccaccio et al., Cell Mol Life Sci, 2014). Also for the plant vacuolar CLC-a transporter, inhibition occurs with an apparent affinity of 12 nM (Carpaneto et al., Sci Rep, 2017). These discrepancies need to be discussed in detail. In addition, a concentration of "~1 mol% PI(3,5)P2 in the cytosolic membrane" appears to be extremely high. Indeed, PI(3,5)P2 is actually a very minor lipid and it is thought that the low level of PI(3,5)P2 keeps channels like TRPML shut. Such channels are only activated in response to a synthesis of PI(3,5)P2. Thus, the hypothesis of a tonic block of CLC-7/Ostm1 that is released by a decrease of PI(3,5)P2, i.e. the idea that CLC-7/Ostm1 activity is relevant only in response to a decrease of PI(3,5)P2 appears rather unphysiological.

Therefore, it is essential to demonstrate at what PI(3,5)P2 concentrations are CLC-7 wt and Y715C mutant inhibited. Are these concentrations physiological (or in a range that we think can be reached under physiological conditions), are they higher or lower than those that inhibit other lysosomal channels, primarily TPC2 and TRPML1, both of which can have the same effect on lysosomal pH. One possible approach for evaluating whether CLC-7 is inhibited by physiological concentrations of PI(3,5)P2 would be to use PM targeted TPC1 for calibration, see e.g. She et al., 2018 (https://doi.org/10.1038/nature26139, extended Figure 2b). The mutant Y715C should also be evaluated.

2) The electrophysiological experiments are not sufficiently documented. In particular, the kinetics of the onset of inhibition by PI(3,5)P2 upon break-in into the whole cell configuration provides important information: is the time course compatible with the assumptions? Is the time course reproducible? How fast is steady state achieved? These are standard, and extremely easy to implement controls of experimental quality that need to be shown.

3) The main rationale for CLC-7 being responsible for the apilimod-induced effects on lysosomal pH and size in Figures1-4 is the lack of apilimod-induced effects on CLC-7-KO cells (shown in Figure 5). Because CLC-7-KO lysosomes could have altered expression of other PI(3,5)P2-sensitive channels (a point that is not considered by the authors and it is not easy to quantify), the authors should use well-characterized inhibitors of these channels to separate the CLC-7-mediated effect from other PI(3,5)P2-sensitive channels. As mentioned in the discussion, CLC-7 is not the only transporter regulated by PI(3,5)P2, but it is the only one reported to be negatively regulated. Two other endolysosomal cation channels, TRPML1 and TPC2 are positively regulated by PI(3,5)P2. Thus, if at rest PI(3,5)P2 concentration is high enough to keep CLC-7 closed, preventing Cl^-^ from being transporter in the lumen, ca^2+^ and/or Na^+^ would also leave the lumen through TRPML1 and TPC2. (For example, the TPC2 KO leads to enlarged melanosomes (lysosomal-related organelles; Ambrosio et al., PNAS 2016), suggesting that TPC2 could be open at baseline.) Although other channels maybe be beyond the scope of this paper, the question of whether endolysosomes can function with a CLC-7 closed at baseline must at least be discussed.

4) One extremely important aspect regarding the role of CLC-7/Ostm1 in lysosomal enlargement, i.e. chloride transport, has been completely ignored by the authors. In fact, chloride accumulation has been proposed to be one of the major functions of CLC-7/Ostm1. This is related to the repeated finding that knock-out of CLC-7/Ostm1 does not impair lysosomal acidification, seen also here by the authors. It is therefore highly desirable to investigate the levels of lysosomal chloride in the various experimental conditions, i.e. PIKFYVE inhibition and CLC-7/Ostm1 knock-out. It is surprising that the authors not even mention this possibility. At minimum, the potential role of chloride, and the related literature, need be discussed as a viable possibility.

5) The observation that cells with CRISPR-mediated CLC-7 KO do not have altered pH, although consistent with the observation in other systems, could reflect the adaptation of the cells to lack of CLC-7 by changing the expression of activity of the other channels and transporters in the membrane. Without being able to acutely downregulate or block CLC-7, it cannot be concluded that it does not regulate luminal pH as baseline.

[Editors’ note: further revisions were suggested prior to acceptance, as described below.]

Thank you for resubmitting your work entitled "Tonic inhibition of the chloride/proton antiporter ClC-7 by PI(3,5)P2 is crucial for lysosomal pH maintenance" for further consideration by *eLife*. Your revised article has been evaluated by Richard Aldrich (Senior Editor) and a Reviewing Editor.

The reviewers are pleased with your revisions and have identified just a few remaining issues that should be straightforward for you to address:

1) We appreciate your rationale for postponing chloride measurements for future work. Nevertheless, given that the function of CLC-7 in accumulation of chloride has been a leading hypothesis in the literature, and that hyperactivity caused e.g. by diminished PIP block or mutation-induced over activity could well be causative for lysosome enlargement, the possibility of chloride accumulation should be discussed.

2) In the TPC1 PI(3,5)P2 calibration experiments, the original TPC1 current traces should be shown.

3) The concentration unit is mM, not mm.

---

## [Author Response]

Essential revisions:1) The concentrations of PI(3,5)P2 used in the patch clamp experiments are very large compared to what are thought to be physiological concentrations, based on studies investigating other PI(3,5)P2-sensitive channels and transporters. For comparison, reported values for activation of lysosomal cation channels for PI(3,5)P2 are three orders of magnitude smaller for mouse TRPML1 (Dong et al., JBC, 20210), and more than two orders of magnitude smaller for Drosophila TRPML1 and human TPC2 (Feng et al., JBC, 2013; Wang et al., Cell, 2012; Boccaccio et al., Cell Mol Life Sci, 2014). Also for the plant vacuolar CLC-a transporter, inhibition occurs with an apparent affinity of 12 nM (Carpaneto et al., Sci Rep, 2017). These discrepancies need to be discussed in detail. In addition, a concentration of "~1 mol% PI(3,5)P2 in the cytosolic membrane" appears to be extremely high. Indeed, PI(3,5)P2 is actually a very minor lipid and it is thought that the low level of PI(3,5)P2 keeps channels like TRPML shut. Such channels are only activated in response to a synthesis of PI(3,5)P2. Thus, the hypothesis of a tonic block of CLC-7/Ostm1 that is released by a decrease of PI(3,5)P2, i.e. the idea that CLC-7/Ostm1 activity is relevant only in response to a decrease of PI(3,5)P2 appears rather unphysiological.Therefore, it is essential to demonstrate at what PI(3,5)P2 concentrations are CLC-7 wt and Y715C mutant inhibited. Are these concentrations physiological (or in a range that we think can be reached under physiological conditions), are they higher or lower than those that inhibit other lysosomal channels, primarily TPC2 and TRPML1, both of which can have the same effect on lysosomal pH. One possible approach for evaluating whether CLC-7 is inhibited by physiological concentrations of PI(3,5)P2 would be to use PM targeted TPC1 for calibration, see e.g. She et al., 2018 (https://doi.org/10.1038/nature26139, extended Figure 2b). The mutant Y715C should also be evaluated.

This is an important point and we performed the suggested experiments to address it. We calibrated our experimental system against the TPC1 results in She et al. 2018 by expressing plasma membrane‐targeted mouse TPC1 (Origene Cat. #MR210795) in HEK‐293 cells and measuring current responses after including a range of concentrations of PI(3,5)P2 diC8 in the pipette solution, from 0 to 100 µM. After allowing solution exchange with the cytosol for 3 min, we measured an EC50 of ~10 µM for mTPC1, whereas She et al. saw a value of ~145 nM in their experimental system. This suggests that our experimental setup requires much higher concentrations of PIP2 to achieve the same effect as others have seen in systems with more direct means to apply the lipid and with more lysosome‐like membrane environments. We suspect that the amount of PI(3,5)P2 available to ClC‐7 in the plasma membrane is significantly less than the solution concentration in the pipette suggests. Note that the literature examples cited above mostly used patch‐clamp of enlarged endolysosomes, with lipid applied in the bath solution and perfused across the vacuoles, a setup that may require lower PIP2 concentrations to elicit the effect. Based on the comparison between CLC‐7 and TPC1 in the plasma membrane, we suspect that the affinity of the transporter for PI(3,5)P2 is substantially higher than our measured EC50 and have noted this in the text.

2) The electrophysiological experiments are not sufficiently documented. In particular, the kinetics of the onset of inhibition by PI(3,5)P2 upon break-in into the whole cell configuration provides important information: is the time course compatible with the assumptions? Is the time course reproducible? How fast is steady state achieved? These are standard, and extremely easy to implement controls of experimental quality that need to be shown.

We apologize to the reviewers for not including these results in the original version and have clarified this issue in our revision. We determined our time point (3 min) for these measurements by analyzing the time course of the CLC‐7 response at a series of times after breaking into the cell. We found that the effect had essentially saturated by the three‐minute point, which is why we reported that time point. We have added a supplemental figure (Figure 6--figure supplement 1) to demonstrate the kinetics of inhibition as requested by the reviewers.

3) The main rationale for CLC-7 being responsible for the apilimod-induced effects on lysosomal pH and size in Figures1-4 is the lack of apilimod-induced effects on CLC-7-KO cells (shown in Figure 5). Because CLC-7-KO lysosomes could have altered expression of other PI(3,5)P2-sensitive channels (a point that is not considered by the authors and it is not easy to quantify), the authors should use well-characterized inhibitors of these channels to separate the CLC-7-mediated effect from other PI(3,5)P2-sensitive channels. As mentioned in the discussion, CLC-7 is not the only transporter regulated by PI(3,5)P2, but it is the only one reported to be negatively regulated. Two other endolysosomal cation channels, TRPML1 and TPC2 are positively regulated by PI(3,5)P2. Thus, if at rest PI(3,5)P2 concentration is high enough to keep CLC-7 closed, preventing Cl^-^ from being transporter in the lumen, ca^2+^ and/or Na^+^ would also leave the lumen through TRPML1 and TPC2. (For example, the TPC2 KO leads to enlarged melanosomes (lysosomal-related organelles; Ambrosio et al., PNAS 2016), suggesting that TPC2 could be open at baseline.) Although other channels maybe be beyond the scope of this paper, the question of whether endolysosomes can function with a CLC-7 closed at baseline must at least be discussed.

This is an excellent point and one which, of course, interests us greatly. We do think that the interplay between all of these ion permeabilities is likely to be very important in lysosomal homeostasis. However, we agree with the reviewers that careful study of these interactions will be an enterprise unto itself and must wait for a further publication. In recognition of the importance of these issues, we have expanded our discussion to address them more directly, including the last point about a ClC‐7 closed at baseline.

4) One extremely important aspect regarding the role of CLC-7/Ostm1 in lysosomal enlargement, i.e. chloride transport, has been completely ignored by the authors. In fact, chloride accumulation has been proposed to be one of the major functions of CLC-7/Ostm1. This is related to the repeated finding that knock-out of CLC-7/Ostm1 does not impair lysosomal acidification, seen also here by the authors. It is therefore highly desirable to investigate the levels of lysosomal chloride in the various experimental conditions, i.e. PIKFYVE inhibition and CLC-7/Ostm1 knock-out. It is surprising that the authors not even mention this possibility. At minimum, the potential role of chloride, and the related literature, need be discussed as a viable possibility.

We have added a paragraph to the discussion addressing the role of [Cl-].

5) The observation that cells with CRISPR-mediated CLC-7 KO do not have altered pH, although consistent with the observation in other systems, could reflect the adaptation of the cells to lack of CLC-7 by changing the expression of activity of the other channels and transporters in the membrane. Without being able to acutely downregulate or block CLC-7, it cannot be concluded that it does not regulate luminal pH as baseline.

We agree with the reviewers here about other possible explanations for the lack of an effect of ClC‐7 KO on baseline pH; indeed we have data suggesting that the situation is indeed complicated. Nevertheless, we prefer not to take on this controversy in this work. As the reviewers point out, without acutely inhibiting ClC7, we can’t make firm conclusions about its role at baseline; we were careful, however, to word our conclusion in a way that leaves open the possibility that baseline ClC‐7 permeability contributes to acidification (“largely inhibited by PI(3,5)P2 in normal cellular conditions..” We also added a sentence to the discussion pointing out that compensatory changes could be responsible for the maintained pH in the KO cells/mice.

[Editors’ note: further revisions were suggested prior to acceptance, as described below.]

The reviewers are pleased with your revisions and have identified just a few remaining issues that should be straightforward for you to address:1) We appreciate your rationale for postponing chloride measurements for future work. Nevertheless, given that the function of CLC-7 in accumulation of chloride has been a leading hypothesis in the literature, and that hyperactivity caused e.g. by diminished PIP block or mutation-induced over activity could well be causative for lysosome enlargement, the possibility of chloride accumulation should be discussed.2) In the TPC1 PI(3,5)P2 calibration experiments, the original TPC1 current traces should be shown.3) The concentration unit is mM, not mm.

We have fulfilled all the requests of the reviewers. We added a paragraph to the discussion addressing the possible effects of increased Cl^-^ flux (middle page 21); we added raw traces to the TPC1 calibration (Figure 6 supplement 3), and we corrected the units mm to mM.